# An Analysis of the Self-Healing and Mechanical Properties as well as Shape Memory of 3D-Printed Surlyn^®^ Nanocomposites Reinforced with Multiwall Carbon Nanotubes

**DOI:** 10.3390/polym15214326

**Published:** 2023-11-04

**Authors:** Rocío Calderón-Villajos, María Sánchez, Adrián Leones, Laura Peponi, Javier Manzano-Santamaría, Antonio Julio López, Alejandro Ureña

**Affiliations:** 1Department of Applied Mathematics, Materials Science and Engineering and Electronic Technology, Universidad Rey Juan Carlos, Calle Tulipán s/n, 28933 Móstoles, Spain; javier.manzano@ciemat.es (J.M.-S.); antoniojulio.lopez@urjc.es (A.J.L.); alejandro.urena@urjc.es (A.U.); 2Instituto de Ciencia y Tecnología de Polímeros, Calle Juan de la Cierva 3, ICTP-CSIC, 28006 Madrid, Spainlpeponi@ictp.csic.es (L.P.)

**Keywords:** 3D printing, carbon nanotubes, self-healing, extrusion, mechanical properties, shape memory

## Abstract

This research work studies the self-healing ability, mechanical properties, and shape memory of the polymer Surlyn^®^ 8940 with and without multiwall carbon nanotubes (MWCNTs) as a nanoreinforcement. This polymer comes from a partially neutralized poly(ethylene-co-methacrylic acid) (EMAA) ionomer copolymer. MWCNTs and the polymer went through a mixing process aimed at achieving an excellent dispersion. Later, an optimized extrusion method was used to produce a uniform reinforced filament, which was the input for the 3D-printing process that was used to create the final test samples. Various concentrations of MWCNTs (0.0, 0.1, 0.5, and 1.0 wt.%) were used to evaluate and compare the mechanical properties, self-healing ability, and shape memory of unreinforced and nanoreinforced materials. Results show an enhancement of the mechanical properties and self-healing ability through the addition of MWCNTs to the matrix of polymer, and the specimens showed shape memory events.

## 1. Introduction

Surlyn^®^ is the trademark name given to EMAA, a copolymer neutralized with sodium salt known as an ionomer [1]. This polymer has been widely studied as it contains excellent self-healing properties [2,3,4]. Although self-healing materials can be an attractive option to increase the damage tolerance of structures, Surlyn^®^ has poor mechanical properties, which can be a limitation for some industrial applications. Increasing the mechanical properties of this polymer without decreasing its self-healing capabilities could expand its applications [5,6]. The use of MWCNTs as reinforcements have been proven in recent years [7,8] due to their low densities and remarkable mechanical properties, making them a potentially suitable nanoreinforcement [9,10]. Moreover, their physical, thermal, and optical properties make them appropriate options for fabricating hybrid nanocomposites with improved mechanical properties. To obtain such good properties (better elastic modulus and increased tensile strength) in the resulting nanocomposite, uniform dispersion and strong interfacial adhesion of the MWCNTs to the polymeric matrix of the original polymer are required [11,12]. The synergistic effect of the high thermal conductivity of MWCNTs (around 3000 W/mK) and the achievement of a good dispersion of the MWCNTs in the matrix of the polymer resulted in a better self-healing ability [13].

A growing field of study is the exploration of methods to provide self-healing properties for polymeric materials. In a range of applications, self-healing materials offer the potential to greatly extend the useful life and safety of structural components. To produce a self-healing process in these materials, an outside stimulus, such as thermal, photonic, or chemical activation, is required. These methods require some polymers capacity to connect or recross their chains [14,15]. In this way, up to 20 mol% of ionic species are typically seen in ionomeric polymers because the ionomer polymers are made into chains containing ionic groups, and they enrich/add the polymer matrix with enough counterions by adjusting the ionic content; an ionomeric polymer with self-healing capabilities can have different properties [16,17].

Fused Deposition Modeling (FDM) Technology is a 3D-printing process used to manufacture novel materials [18]. Among its advantages are faster production and the freedom to customize the shapes and geometry of the produced samples at an affordable cost [19]. This technology is applied to a wide range of applications [20].

There are polymeric smart materials that can revert to their original permanent shape from a distorted condition when exposed to external stimuli. This phenomenon is known as shape-memory polymers (SMPs) [21]. Thanks to the innovative combination of SMPs, FDM, also known as 3D-printing, has become a growing technology to produce more inventive devices and possible practical applications in biomedical, electronics, and sensors [22,23]. The use of SMPs in 3D printing has been hindered, nevertheless, by limited printable materials and the inadequate mechanical properties and thermomechanical shape memory of currently available 3D-printing materials [24,25].

The main goal of this research is to produce a novel nanocomposite polymer reinforced with MWCNTs, with both self-healing and shape memory properties obtained via a 3D-printing process.

## 2. Materials and Experimental Procedure

### 2.1. Materials

The polymer used to produce the nanocomposites was EMAA copolymer neutralized with 30 wt.% sodium under the commercial name of Surlyn 8940^®^, manufactured by DuPont (Wilmington, DE, USA). Its density is 95 g/cm^3^, its melt flow index is 2.8 g/10 min, and the melting point measured by DSC is 94 °C. The nanoreinforcement used (the MWCNTs) was under the commercial name NC7000, manufactured by Nanocyl (Sambreville, Belgium), and it presented an average diameter of 9.5 nm and length of 1.5 µm.

### 2.2. Extruder Filament Preparation

Producing a filament with a constant diameter from the starting material is critical in order to obtain high-quality specimens. The resulting filament will be used afterward for 3D printing the samples with the required quality by means of Fused Deposition Modeling (FMD). Four different filaments were produced using Next 1.0 3DEVO extruder (Utrech, The Netherlands). One of the filaments was made of raw Surlyn^®^, while the other three were made of nanoreinforced Surlyn^®^ with different concentrations of MWCNTs (0.1, 0.5, and 1 wt.%). The feedstock material of the filament extruder needed to be in the form of small pellets. For plain Surlyn^®^ filament, Surlyn^®^ pellets were directly used as base material to produce the filaments. In the cases of nanoreinforced filaments, the Surlyn^®^ pellets were mixed with the corresponding concentrations of MWCNTs by using a Brabender^®^ mixer (Haake Rheocord 9000, Waltham, MA, USA). The process consisted of melting the polymer at 120 °C and then introducing the corresponding wt.% MWCNTs into a hopper for 10 min. The resulting mixture was taken and broken back into small pellets. The configuration of the extruder was performed to obtain high-quality filaments with a diameter of 1.75 mm. To achieve this, the extrusion speed was set to 4.2 rpm, and the temperature of the different extruder’s sections were set to 165 °C, 180 °C, 185 °C, and 165 °C, respectively. The final nozzle had good precision to yield filaments of the desired section, so it was configured to 1.75 mm.

### 2.3. 3D-Printing of Surlyn^®^ Using FDM

The previously extruded Surlyn^®^ filaments were used to feed a Witbox printer from BQ company (Utrech, The Netherlands) that was used to print the samples with different concentrations of MWCNTs (0.0, 0.1, 0.5, and 1.0 wt.%) by means of the FDM technology. Table 1 collects the printing parameters used to manufacture the specimens. The pattern of the samples pursues the ISO-527-2-2012 for tensile test samples (35 × 2 × 1 mm dimension) [26], and the g-code file was created by slicing a file using Ultimaker Cura 4.6 software. Table 1 shows the parameters used for the 3D printing of the samples, which had been previously optimized.

### 2.4. Self-Healing Testing

The self-healing properties of Surlyn^®^ are well-known and studied; the objective is to compare the properties of the 3D-printed raw Surlyn^®^ samples versus nanoreinforced Surlyn^®^ samples. The test procedure consisted of applying a constant load of 10 kg for 15 s by means of Shore D standard ASTM D2240-05 [27] to produce indentation damages to the surface of the specimens. The metric used to evaluate the self-healing ability of the samples is called volumetric recovery percentage (V %), and it is defined in Equation (1).
(1)$$V(\%)=\frac{\left|V_{f}-V_{o}\right|}{V_{o}}\times 100$$
where V_o_ is the initial volume of the indentation damage, and V_f_ is the final volume after heating in an oven at 80 °C for 1 h to perform the self-healing process. Temperature and heating duration were chosen based on previous research [13,28]. The different volumes were measured using a 3D optical profiler from Zeta Instruments (Zeta-20 model, Madrid, Spain), which generated 3D micrographs. These micrographs were obtained before and after the recovery process and processed afterward with the Mountain Map Premium 7.1 software to obtain the volumetric recovery percentage.

### 2.5. Mechanical Testing

A sample size of 35 × 2 × 1 mm dimension was tested using a Zwick machine (Madrid, España) with a load cell of 20 N and a speed of 10 mm/min at 25 °C. Four samples were tested in each case.

### 2.6. SEM Study: Dispersion Analysis of the Nanoreinforcement

To study the dispersion of the MWCNTs along the 3D-printed specimens, the samples were cut with a blade and, afterward, bent until the specimen breakage was produced. Then, the sectioned surface was sputter-coated with a golden layer of about 7.7 nm of thickness to make it conductive. A scanning electron microscope (Nova Nano SEM230, apparatus from Philips, installed at Centro de Apoyo Tecnológico in Universidad Rey Juan Carlos, Madrid, Spain) at an accelerating voltage of 7.5 kV was used to take images of the fractured section of the samples.

### 2.7. Thermal Analysis

The thermal properties were investigated by Differential Scanning Calorimetry (DSC) analysis performed in a Mettler Toledo DSC822e instrument (Madrid, Spain) under nitrogen flow (30 mL/min) by sealing the samples (about 10 mg) in aluminum pans.

Thermal cycles were composed by the following “heat/cool/heat” procedure: heating at 10 °C min^−1^ from 0 °C to 150 °C, cooling at 10 °C min^−1^ to 0 °C, and heating again at 10 °C min^−1^ to 150 °C. The first scan was to erase the thermal history of the processed samples. In our case, we prefer to refer to this thermogram in order to establish the right parameters for the thermo-mechanical shape memory cycles. From the second and the third scans, the crystallization temperatures (T_c_), the melting temperatures (T_m_), and the melting enthalpy (ΔH_m_) were obtained, respectively. The degree of crystallinity (X_c_) of each 3D-printed sample was calculated in accordance with the equation below:(2)$$\chi_{c}\left(\%\right)=\left(\frac{{\Delta H}_{m}}{{\Delta H}_{m}^{100}}\right)\times 100$$
where ∆H_m_^100^ is the specific melting enthalpy for a 100% crystalline PE (278 J/g) [29].

### 2.8. Thermally Activated Shape Memory Properties

Thermally activated shape memory characterization was performed by thermo-mechanical cycles by using a stress-controlled DMA Q800 from TA Instruments (Madrid, Spain) in tension mode. Samples for the thermo-mechanical cycles were printed directly by FDM 3D printer. Both dual- and triple-shape memory behaviors were investigated in this study, properly identifying the programming and recovery steps for the thermo-mechanical cycles. In particular, for the dual-shape memory experiments, the samples were heated at the switching temperature (T_sw_) for 5 min, in this case, 60 °C. Then, stress-controlled uniaxial stretching was applied until a fixed percentage of deformation, i.e., 50%, our ideal temporary shape, was reached, followed by cooling at 0 °C (T_fix_) under the same constant stress, which was released after 10 min in order to fix the temporary shape of the sample. Finally, once again applying the T_sw_ at 3 °C min^−1^, a free-strain recovery was obtained, recovering the initial shape of the sample. The thermo-mechanical cycles were repeated for 4 times. Figure 1a schematically reports a thermo-mechanical cycle as well as the sample evolution during the fix and recovery stage of the shape memory cycles. In Figure 1b, the schematic representation of the thermo-mechanical cycle for the study of the triple-shape memory properties is reported, where a triple-shape memory response indicated that 2 different temporary shapes and 2 different T_sw_ are involved in the experiment—in this case, 60 and 80 °C, respectively.

Therefore, for studying the thermally induced triple-shape memory effect, samples were heated at the T_sw1_ for 5 min and were uniaxially stretched under controlled stress until 40% deformation was achieved in order to fix the first temporary shape. After that, the samples were cooled down for 10 min at the T_sw2_, and by unloading the stress for 10 min, temporary shape 2 was fixed. In order to program the third shape, stress-controlled uniaxial stretching was applied again at the same temperature until 80% deformation was achieved, the 2nd ideal temporary shape. After cooling down the sample at the T_fix_ under the same constant stress for 10 min, the stress was removed again. Finally, a free-strain recovery was obtained when heating again at the previous switching temperatures, T_sw2_ and T_sw1_, respectively, heating at 3 °C min^−1^. In order to quantitatively the shape memory response of the material, the strain fixity ratio as well as the strain recovery ratio have to be calculated, following the Equations (3) and (4), respectively:(3)$$R_{f}\left(N\right)=\frac{\varepsilon_{u}\left(N\right)}{\varepsilon_{m}}\times 100\%$$
(4)$$R_{r}\left(N\right)=\frac{\left(\varepsilon_{m}-\varepsilon_{p}\left(N\right)\right)}{\varepsilon_{m}-\varepsilon_{p}\left(N-1\right)}\times 100\%$$
where N indicates the number of cycles, Ɛ_m_ is the deformed strain, Ɛ_u_ is the fixed strain, and Ɛ_p_ is the recovered strain [30], as also indicated in Figure 1.

## 3. Results and Discussion

### 3.1. Extruded Polymer Filaments and Tensile Test of the 3D-Printed Specimens

The extrusion parameters were optimized to obtain high-quality filaments with a constant diameter of 1.75 mm without either heterogeneities or occluded air inside the filaments. Figure 2a shows all filaments obtained; the transparent one is the filament manufactured from pellets of raw Surlyn^®^, and the black ones correspond to the 0.1, 0.5, and 1.0 wt.% MWCNTs of nanoreinforced filament, respectively. Figure 2b shows, from left to right, the nanoreinforced specimens: 0.0, 0.1, 0.5, and 1.0 wt.% MWCNTs. In these latter filaments, no color difference was observed among them; all nanoreinforced filaments are completely black without any color difference visible to the naked eye. The color homogeneity also demonstrates a good dispersion of the nanoreinforcements in the filaments in all cases.

Figure 2b shows 3D-printed specimens (35 × 2 × 1 mm dimension,) used for the tensile test. There is an observable, homogenous black color in all the nanoreinforced samples; therefore, a good mixture of the MWCNTs was also obtained (Figure 2b).

### 3.2. Self-Healing Ability

The self-healing process was induced by performing several indentations using the Shore D standard on the surface of unreinforced and nanoreinforced Surlyn^®^ specimens and then heating the samples, as described in Section 2.4: Self-healing testing. Table 2 shows the volumetric recovery percentage of all specimens where it can be observed that self-healing occurred in all cases, and the self-healing ability of the polymer was not changed despite the sample fabrication process. An increase in self-healing ability is appreciated as the wt.% of MWCNTs increases [11,31]. The volumetric recovery percentage is similar between raw Surlyn^®^ and the sample with a lower concentration of nanoreinforcement (0.1 wt.%). However, samples with higher concentrations of MWCNTs showed an increase in self-healing ability [2,32]. The differences in the results of the self-healing phenomena are due to the synergistic effect of the high thermal conductivity of MWCNTs (around 3000 W/mK) and the achievement of a good dispersion of the MWCNTs in the matrix of the polymer [13], which increases the heat conductivity throughout the polymer and, therefore, leads to an improvement in the observed volumetric recovery percentage.

Figure 3 shows how the material heals the induced damage caused by the indentation, decreasing the depth of the damage from the center part of the hole. The blue color in the images, representing the maximum penetration zones (in the middle of the residual print), decreases after the thermal healing treatment.

Figure 3 compares the damages before and after the recovery process. Samples were exposed to convection heat in an oven at 80 °C for 60 min. The hole closure or elastic response of the Surlyn^®^ ionomer is clearly evident in all cases. The Surlyn^®^ polymer has the ability to self-heal due to the ion-hopping mechanism and the elastic movement of the polymeric chains, which are responsible for the self-healing process described in [28].

### 3.3. Printed Tensile Test 3D-Printed Specimens

Figure 4 shows the stress–strain curves from the tensile test of raw Surlyn^®^ and nanoreinforced Surlyn^®^ with different concentrations of MWCNTs (0.1, 0.5, and 1 wt.%). The analysis of these curves provides the mechanical properties summarized in Table 2.

The samples with nanoreinforcement show an improvement in the mechanical properties over raw Surlyn^®^ samples. This improvement indicates a good distribution of the nanoreinforcements inside the polymer matrix. The addition of MWCNTs increases the tensile strength, and this improvement is slightly higher as the concentration of MWCNTs is raised. The tensile strength of the unreinforced sample was lower than the reference value from the Dupont^®^ datasheet. This result was expected, as the mechanical properties of the samples are heavily affected by the manufacturing process. The way in which the internal structure of the samples is formed on the meso-scale is different in 3D-printed samples [33] compared to the structure present in the bulk material. Figure 5 shows the triangular meso-structures (interbead voids) formed during the FMD 3D-printing process of the nanoreinforced sample. A good contact area between the printed tracks is very important to obtain the desired properties on the meso-scale. Potential gaps between tracks during the 3D-printing process can be minimized by defining a small overlap between the tracks. This parameter is configurable, and it was optimized for this type of sample [34,35]. Figure 5 shows that the number of triangular meso-structures is low, which is needed to obtain good mechanical properties. Table 2 also shows the value of Young’s modulus of the 3D-printed specimens, and it can be observed that there are no significant variations from Dupont’s reference values, which indicates that there were no degradations of the polymer during the thermal procedures to obtain the final specimens. Regarding elongation, a decrease of around 50% is observed in all cases. In addition, this side effect happens to 3D-printed thermoplastic polymer samples. The manufacturing process consists of stacking a series of discrete layers on top of each other, which decreases the overall cohesion of the sample and, therefore, decreases the measured elongation (the same happens with the tensile strength, as previously described). The addition of MWCNTs to the polymer matrix improves the tensile strength of the samples; specifically, it has improved by 41, 49, and 52% as the studied concentrations increase regarding the raw Surlyn^®^ samples, which helps avoid this undesirable effect. However, as seen in Table 2, it has no effect on the elongation of the samples.

### 3.4. Nano-Structure Analysis and Fracture Characterization

Sample images from the results of the SEM study are shown in Figure 6. The dispersion of MWCNT nanoreinforcements can be observed for all concentrations under the study of (0.1, 0.5, and 1 wt.%), and, in all cases, it shows a good dispersion of MWCNTs in the fracture surface of the polymer. Figure 6a,b show the MWCNTs protruding out of the fracture surface in a sample with a concentration of 0.1 wt.% of MWCNTs. As can be seen, the distribution of MWCNTs is homogeneous, and there is a uniform distribution of the MWCNTs, with no witnessed agglomeration of MWCNTs in the matrix. This is the result of a good interfacial interaction between MWCNTs and polymer chains [36]. Figure 6c,d show the fracture surface of the 0.5 wt.% MWCNT specimens where the MWCNTs are entangled with each other, preserving the initial length and tubular structure after the mixture process, which is an indicator of good adhesion to the matrix. Figure 6e shows the fracture surface on the 1 wt.% MWCNT specimen, the homogeneous dispersion of the MWCNTs along the matrix, as well as an absence of agglomeration, which can be seen on it. Figure 6f shows the fracture surface of 0.5 wt.% of MWCNTs 3D-printed sample after the tensile test (which was carried out until the sample broke). Many of the MWCNTs were pulled out when the fracture happened, so their long length and entangled arrangement can be seen. This also indicates that the MWCNTs are evenly dispersed along the specimen, with good adhesion to the polymeric matrix and preservation of their initial lengths after the tensile test [37]. These observations are aligned with the measured improvements in the mechanical properties of the specimens.

### 3.5. Thermally Activated Dual- and Triple-Shape Memory Effect

As expected, ionomeric resins based on EMAA copolymers such as Surlyn^®^ show a complex structure containing polyethylene (PE) crystals, amorphous polymer chains, and ionic aggregates [30] reflected in their thermal analysis, which must be realized for the following shape memory studies. In particular, in Figure 7, the first heating, cooling, and second heating are reported. The first heating is, thus, reported with consideration that the shape memory has been studied on the samples as obtained from the 3D-printing process without any further thermal treatment. However, from a thermal point of view, two endothermic peaks can be observed. The highest one is associated with the melting temperature of the PE crystals, at about 80 °C, while the first one is the subject of controversial discussion in the scientific literature. From one perspective, it is identified as the melting temperature of small secondary PE crystals [38] due to the annealing effect of the sample at room temperature [39,40,41], and from the other perspective, some authors attribute this endothermic peak to an orden–desorden transition within the ionomeric aggregates [37,42,43,44,45]. Based on our previous studies on Surlyn^®^-based nanocomposites, we consider the first endothermic peak due to the PE secondary crystals, which is in accordance with our previous work [46,47].

In particular, the primary crystals melt at between 70 and 105 °C, with a maximum melting temperature of around 90 °C. The small secondary crystals melt between 45 and 60 °C. The presence of these two different crystalline phases, as well as the different melting intervals, are important as they will be used in following shape memory studies to determine fix and switching phases as well as the switching temperatures. In Table 3, the main thermal parameters in terms of the glass transition temperature, melting temperature, and degree of crystallinity are reported. They calculate for both neat and MWCNT-reinforced 3D samples from the second heating. It is possible to see that no significant variation is presented between the different 3D-printed materials. This fact will be used to design the programming and recovering step of the thermo-mechanical cycles of both dual- and triple-shape memory effects.

Therefore, both a dual- and triple-shape memory study has been performed on the 3D-printed Surlyn^®^-based materials, starting with the dual one. In Figure 8, the dual-shape memory effect on the Surlyn^®^-based 3D-printed materials at 50% deformation is reported. As indicated before and based on our previous study [46,47], a T_sw_ of 60 °C has been considered, being that the PE secondary crystals are the switching domains. All the materials presented a very good shape memory behavior. Moreover, by analyzing the applied stress, we can also confirm the reinforced effect of the MWCNTs on the Surlyn^®^ matrix. In fact, in the case of the neat matrix, the smallest stress is needed to deform the sample, about 5 MPa, while in the case of nanocomposites, an applied stress of about 7 MPa is needed to deform the samples. Moreover, the behavior of the samples, reinforced with and without MWCNTs, are the same in every thermo-mechanical cycle, indicating the good shape memory capability of the 3D-printed samples.

Table 4 reports the R_f_ and R_r_ obtained for the dual-shape memory response of the 3D-printed samples.

From Table 4, it is possible to point out the very good capability of 3D-printed Surlyn^®^-based materials, evidencing an optimal strain fixity ratio higher than 94% in both neat and MWCNT-nanoreinforced materials in all three thermo-mechanical cycles, indicating their optimal capability for fixing their temporary shape. The strain recovery ratio presents values very close to 90%, evidencing the good capability of the 3D-printed samples to recover their initial shape. However, based on our previous study [47], we can check the triple-shape memory effect on Surlyn^®^-based materials; therefore, Figure 9 reports the triple-shape memory responses for both 3D-printed neat and MWCNTs-reinforced Surlyn^®^. Both 3D diagrams and the time evolution of the thermo-mechanical cycles are reported for each sample. In this case, we have two switching temperatures, that is 60 and 80 °C, as well as two temporary shapes at 40% and 80%, respectively.

However, in order to better understand the triple-shape memory effect, we have to consider that with the N shape memory effect, we have an N-1 temporary shape and N-1 switching temperature. Therefore, in our case, we have expected the first switching temperature as 80 °C and the first 40% deformation to fix the first temporary shape, and then the second switching temperature of 60 °C and 80% deformation to fix the second temporary shape. Moreover, in Table 5, the values of the strain fixity ratio as well as of the strain recovery ratio for the three thermo-mechanical cycles for both neat and MWCNTs-reinforced 3D-printed Surlyn^®^ samples are reported.

At the same time, for the dual-shape memory effect, as well as for the triple SME, we have the programming and the recovery stage. During the programming phase, two different temporary shapes have been fixed—in our case, e_u1_ and e_u2_. We consider e_u_ and not e_m_ because we are not in an ideal system in which, once we fix the temporary shape and release the stress, the sample will change in shape. In our case, there is evidence of this fact, and the strain fixity ratio of the first step is not very high. A longer time may be needed to better fix the first temporary shape at 80 °C. Once the first step is performed, we are able to study the second step of the triple-shape memory effect, and then the sample is cooled down to the second switching temperature, which is 60 °C, with a higher 80% deformation. In this case, R_f_ presents very good values, indicating the very good capability to show shape memory at 60 °C. These results confirm the previous ones obtained for the dual-shape memory effect. Once both shapes are programmed, the recovery stage is started, recovering the first shape at T_sw2_, which, again, requires the application of a temperature of 60 °C; then, the 3D-printed sample is able to recover its second, fixed shape, which is 40%. Then, after cooling down the sample, it recovers its initial shape. In both cases, the strain recovery ratios are quite high, at about 90% during the recovery of the second shape and higher than 60% when the first shape is recovered.

## 4. Conclusions

This research work demonstrates that Surlyn^®^ with and without MWCNT-nanoreinforced specimens can be manufactured using 3D-printing technology (FDM) without losing its functional smart properties, such as self-healing and the shape memory effect. The mechanical properties of the obtained specimens improved as the concentration of MWCNTs increased for all concentrations studied. The tensile strength also improved at higher concentrations, which demonstrated that there was a uniform distribution of the MWCNTs, with no witnessed agglomeration of the nanoreinforcement in the matrix. The value of Young’s modulus showed that there were no significant variations from Dupont’s reference values, which indicates no degradations on the polymer. A decrease in elongation was observed in all cases because the layer-by-layer 3D-printed process is known to decrease the overall cohesion of the sample compared to conventional mold-based manufacturing. Moreover, there was an observed improvement in the self-healing values when 0.5 and 1 wt.% MWCNTs are introduced for nanoreinforcement in the matrix, which can be explained by an increase in the thermal conductivity of the nanocomposites. Finally, the thermally activated shape memory effect has been studied; both dual- and triple-shape memory effects have been confirmed in 3D-printed Surlyn^®^-based materials, with very good fixity as well as recovery behavior, indicating that it is possible to process these types of smart polymers via 3D FDM without losing their multifunctionality in terms of their self-healing and shape memory behaviors.

## Figures and Tables

**Figure 1 polymers-15-04326-f001:**
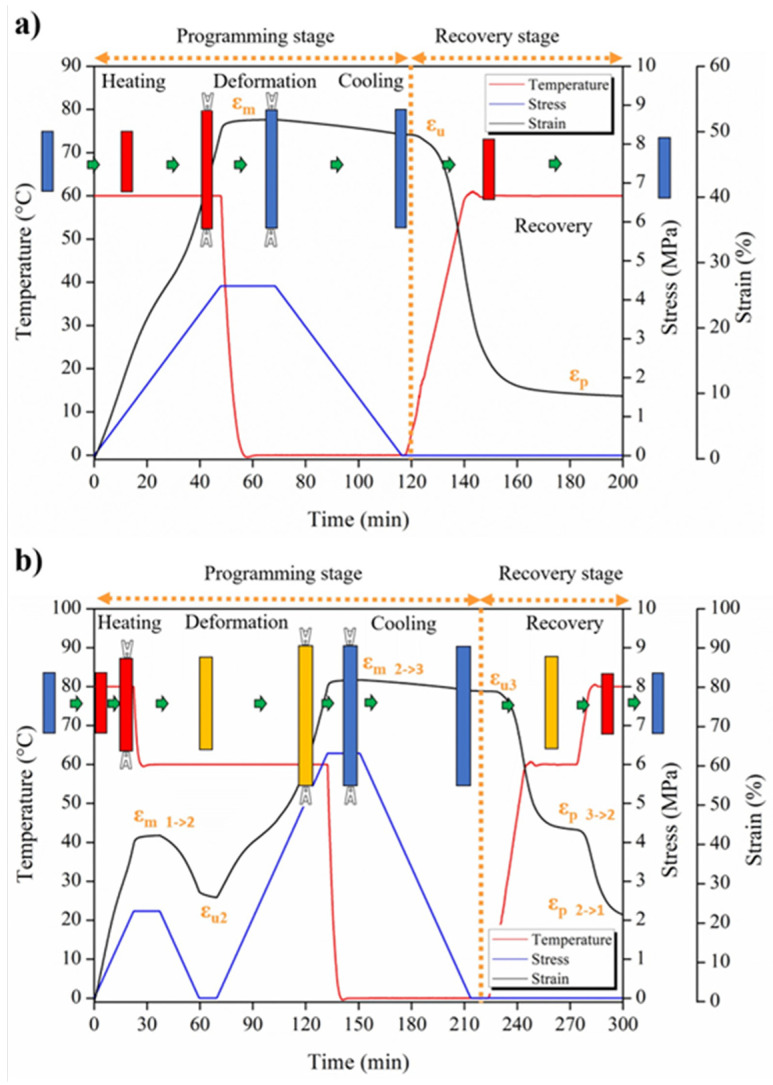
Schematic description of the thermo-mechanical cycle for studying the dual-shape memory (**a**) and the triple-shape memory effect (**b**).

**Figure 2 polymers-15-04326-f002:**
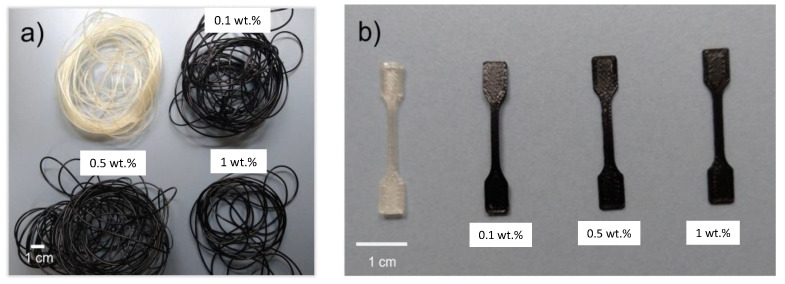
Macroscopic images of (**a**) unreinforced filament and 0.1, 0.5, 1.0 wt.% MWCNTs of the nanoreinforced filament; (**b**) unreinforced specimens and 0.1, 0.5, 1.0 wt.% MWCNTs of the manufactured Surlyn^®^ nanoreinforced specimens.

**Figure 3 polymers-15-04326-f003:**
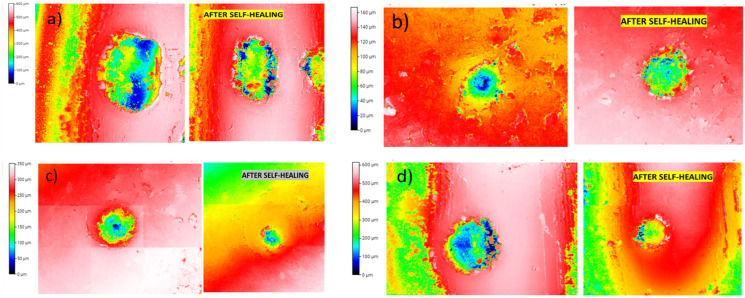
Three-dimensional profilometer micrographs of the generated damage using Shore D standard in Surlyn^®^ specimens: (**a**) raw Surlyn^®^, (**b**) 0.1, (**c**) 0.5, and (**d**) 1 wt.% MWCNTs. Images on the left were taken before the recovery process, and images on the right were taken after the recovery process.

**Figure 4 polymers-15-04326-f004:**
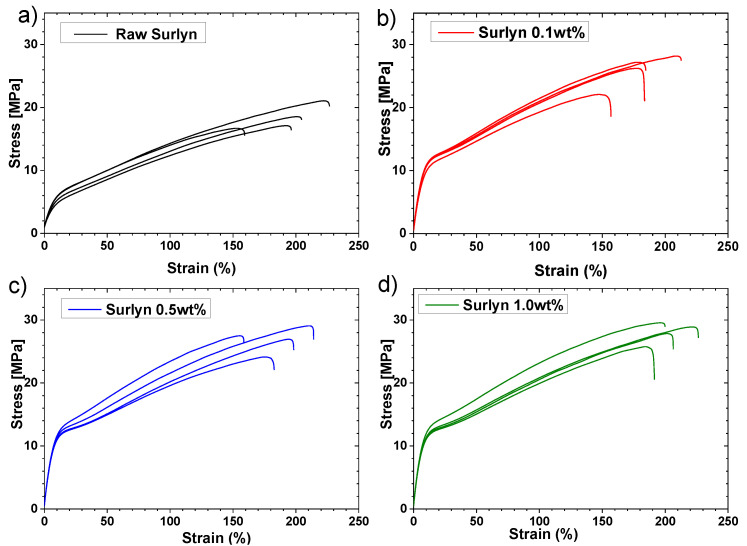
Stress–strain curves of (**a**) unreinforced Surlyn^®^ specimen, (**b**) 0.1, (**c**) 0.5, and (**d**) 1 wt.% of MWCNTs Surlyn^®^ specimens.

**Figure 5 polymers-15-04326-f005:**
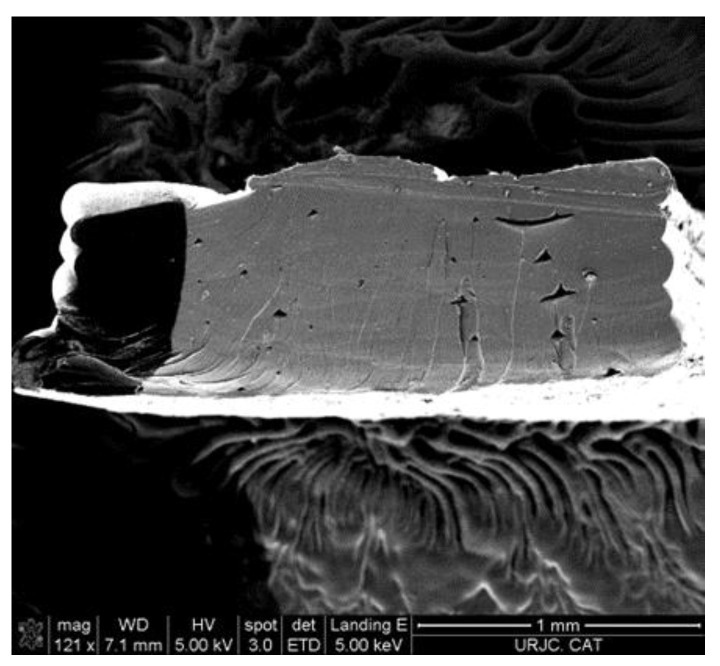
Micrographof the cross-section of 1 wt.% of MWCNTs Surlyn^®^ 3D-printed specimen.

**Figure 6 polymers-15-04326-f006:**
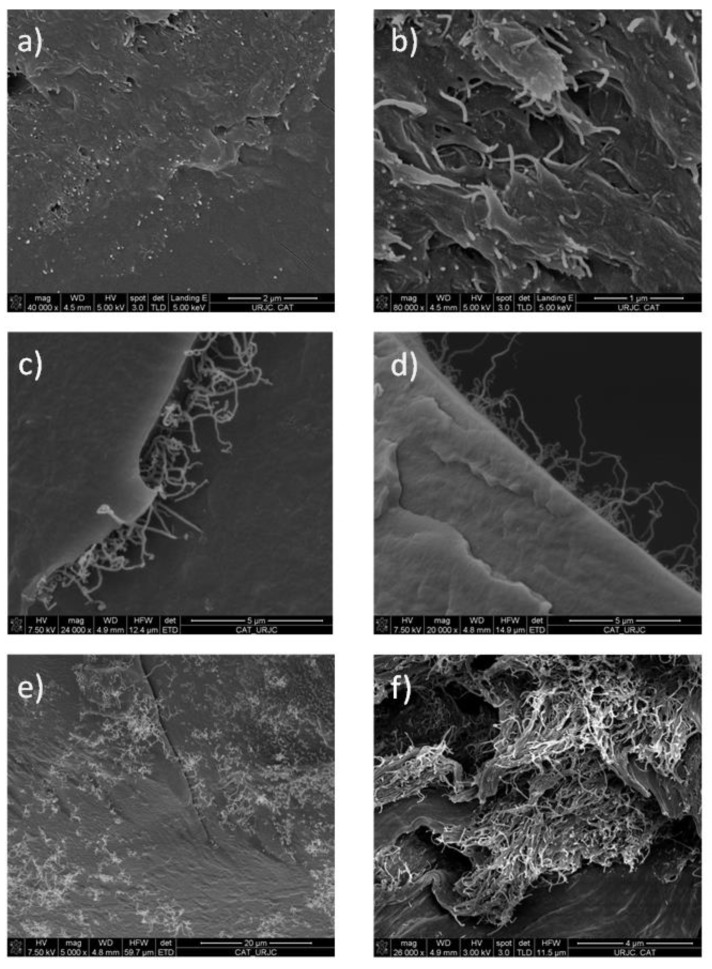
The fracture surface of the Surlyn^®^ 3D-printed specimen: (**a**,**b**) 0.1, (**c**,**d**) 0.5, (**e**) 1 wt.% MWCNTs, and (**f**) 0.5 wt.% MWCNTs after the tensile test.

**Figure 7 polymers-15-04326-f007:**
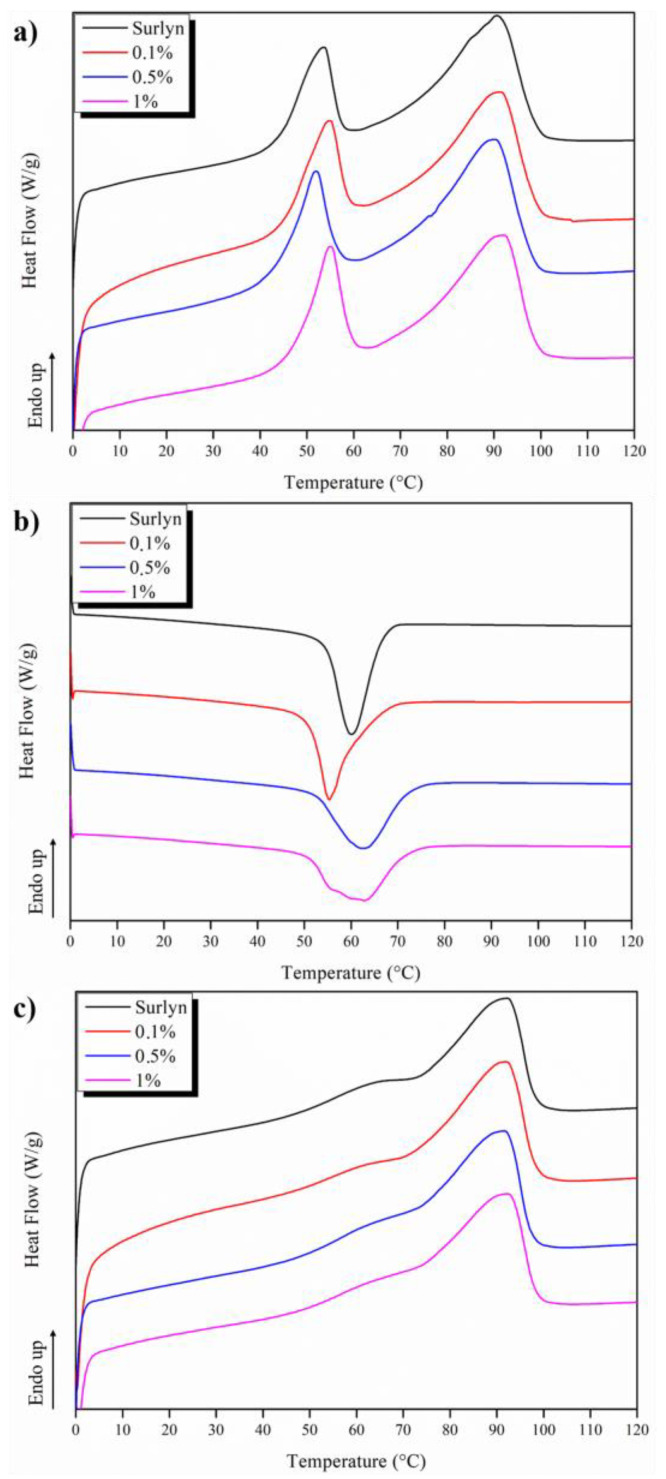
Heating (**a**) cooling, (**b**)–heating, (**c**) scans of DSC analyses for all the 3D-printed materials based on Surlyn^®^.

**Figure 8 polymers-15-04326-f008:**
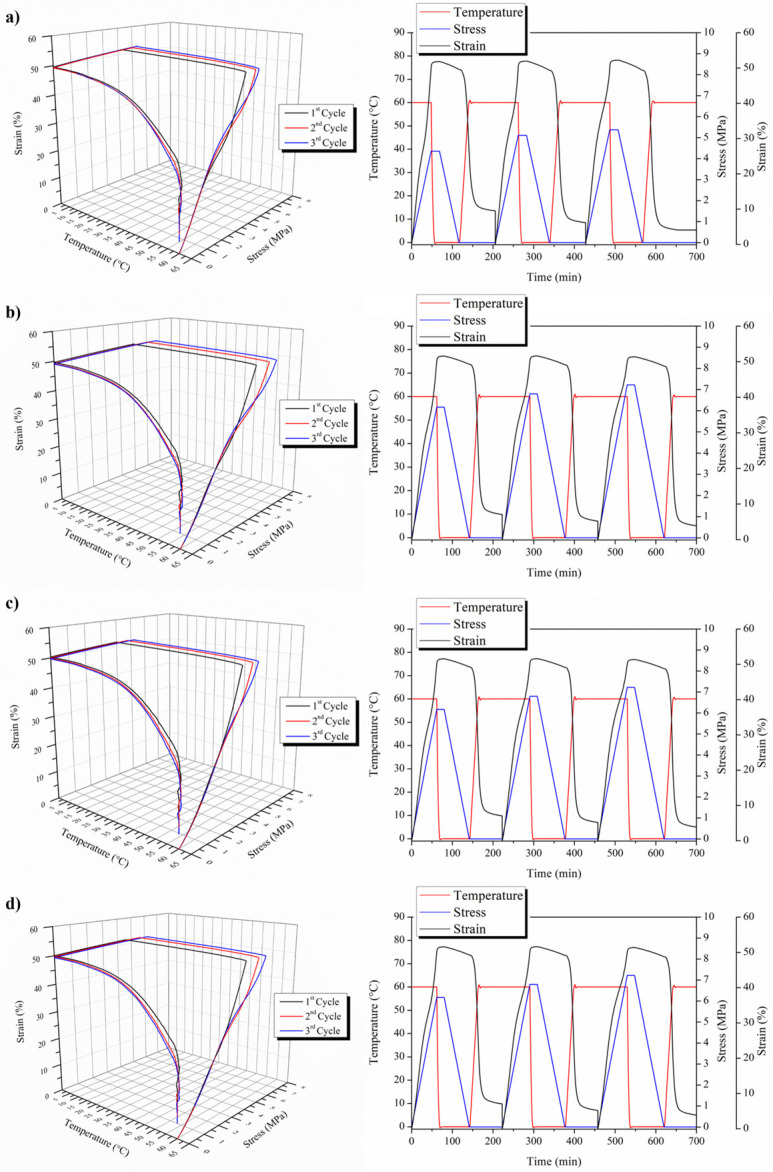
Dual-shape memory effect of both neat and MWCNT-nanoreinforced 3D-printed Surlyn^®^ specimens: (**a**) raw Surlyn^®^,(**b**) Surlyn^®^ +0.1% MWCNTs, (**c**) Surlyn^®^ +0.5% MWCNTs and (**d**) Surlyn^®^ +1% MWCNTs.

**Figure 9 polymers-15-04326-f009:**
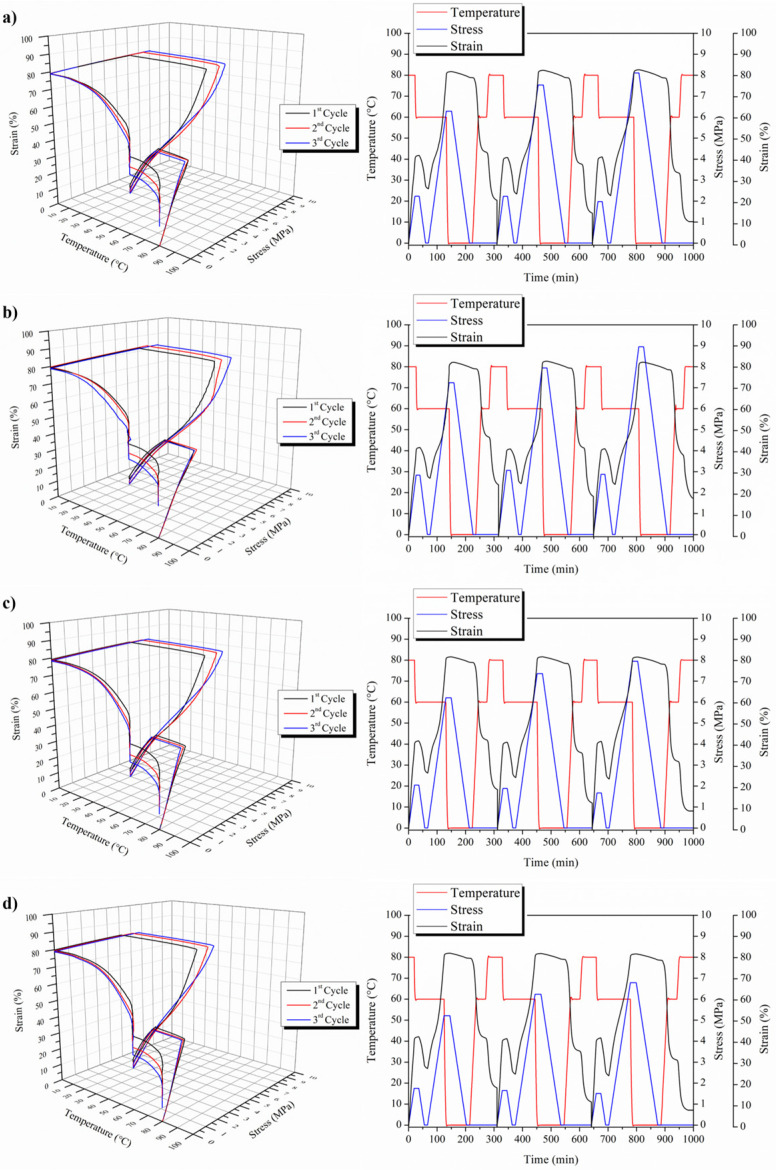
Triple-shape memory effect of both neat and MWCNTs-nanoreinforced 3D-printed Surlyn^®^ samples: (**a**) raw Surlyn^®^, (**b**) Surlyn^®^ +0.1% MWCNTs, (**c**) Surlyn^®^ +0.5% MWCNTs and (**d**) Surlyn^®^ +1% MWCNTs.

**Table 1 polymers-15-04326-t001:** Printing parameters of used FDM 3D to print the specimens.

Material	Surlyn^®^ and Surlyn^®^ + MWCNTs
Infill density	100%
Layer thickness	0.20 mm
Raster orientation or infill orientations	0°/90° central zones 45°/−45° clamping zones
Printing speed	25 mm/s
Extruder temperature (hot-end temperature)	210 °C
Printing bed temperature	Room temperature
Flow	100%

**Table 2 polymers-15-04326-t002:** Average mechanical values of the tested samples.

Specimen	E (MPa)	σ_TS_ (MPa)	Ɛ (%)	V (%)
Dupont datasheet/Hot plate manufactured	290–300 *	15–33 *	470 *	68 ± 5 **
Surlyn^®^	349 ± 34	18 ± 2	200 ± 20	70 ± 5
Surlyn^®^ + 0.1% MWCNTs	324 ± 15	26 ± 2	180 ± 20	66 ± 4
Surlyn^®^ + 0.5% MWCNTs	335 ± 11	27 ± 2	190 ± 20	86 ± 4
Surlyn^®^ + 1% MWCNTs	292 ± 2	28 ± 1	210 ± 10	87 ± 3

E: Young’s modulus. σ_TS_: Tensile strength. ε: Elongation. V (%): volumetric recovery percentage. * Dupont datasheet. ** Hot-plate manufactured material from pellets.

**Table 3 polymers-15-04326-t003:** Thermal behavior of the 3D-printed neat as well as MWCNT-nanoreinforced Surlyn^®^-based materials.

Specimen	T_g_ (°C)	T_m_ (°C)	ΔH_m_ (J/g)	Xc (%)
Surlyn^®^	55	92	24.50	8.8
Surlyn^®^ + 0.1% MWCNTs	55	92	28.05	10.1
Surlyn^®^ + 0.5% MWCNTs	57	92	24.40	8.7
Surlyn^®^ + 1% MWCNTs	57	93	23.66	8.5

**Table 4 polymers-15-04326-t004:** Strain fixity and strain recovery ratios of the dual-shape memory effect for the 3D-printed samples.

T: 60 °C	*R_f_ (%)*			*R_r_ (%)*		
Specimen	1 Cycle	2 Cycle	3 Cycle	1 Cycle	2 Cycle	3 Cycle
Surlyn^®^	96	95	94	81	88	88
Surlyn^®^ + 0.1% MWCNTs	96	95	95	86	90	90
Surlyn^®^ + 0.5% MWCNTs	96	95	94	86	89	89
Surlyn^®^ + 1% MWCNTs	97	95	95	86	90	90

**Table 5 polymers-15-04326-t005:** Strain fixity ratio and strain recovery ratio for the triple-shape memory effect.

T: 60 °C►80 °C	*R*_ƒ_ 1► 2			*R*_ƒ_ 2► 3			*R_r_* 3► 2			*R_r_* 2► 1		
Specimen	1 Cycle	2 Cycle	3 Cycle	1 Cycle	2 Cycle	3 Cycle	1 Cycle	2 Cycle	3 Cycle	1 Cycle	2 Cycle	3 Cycle
Surlyn^®^	63	59	57	96	96	95	84	84	94	51	59	70
Surlyn^®^ + 0.1% MWCNTs	67	61	61	96	96	96	87	97	98	70	71	72
Surlyn^®^ + 0.5% MWCNTs	63	58	57	97	97	97	86	91	93	70	70	73
Surlyn^®^ + 1% MWCNTs	64	61	60	97	96	97	87	92	94	60	61	64

## Data Availability

Data are available upon request.

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
