# Peer review of "An Analysis of the Self-Healing and Mechanical Properties as well as Shape Memory of 3D-Printed Surlyn® Nanocomposites Reinforced with Multiwall Carbon Nanotubes"

_polymers, 2023, doi:10.3390/polym15214326_

Round 1

Reviewer 1 Report

Comments and Suggestions for Authors

This study developed a composite polymer reinforced with MWCNTs, which has better mechanical properties and self-healing ability than raw Surlyn®, by means of a 3D printing process. The experimental data appear to be scientifically sound and the manuscript is overall well written. It is recommended for publication pending minor revisions below.

(1)   The mechanism of the self-healing and recyclability of the as-prepared elastomers needs to be clarified. What is the driving force of the self-healing? Is their any crosslinking agent in the Surlyn® ionomer copolymer?  

(2)   Why did the addition of MWCNTs enhance the self-healing ability of the Surlyn® composites?

(3)   Do the resulted nanocomposites show stickiness? If so, how does the stickiness influence the 3D-printing of the materials?   

(4)  Recent literatures related to the self-healing materials with high mechanical strength, such as Nat. Commun. 2023, 14, 814, can be added to enrich the background of this work.

Comments on the Quality of English Language

It is fine. 

Reviewer 2 Report

Comments and Suggestions for Authors

An analysis of the self-healing and mechanical properties of 3D-printed Surlyn® nanocomposites reinforced with multiwall carbon nanotubes is interesting.

The work presented here is exciting but requires major revisions before it can be accepted for publication. My comments are as follows:

1. The authors need to modify the introduction part. (such as details about the back ground of how MWCNTs increase the mechanical properties, self-healing properties, the importance of self-healing, the role of MWCNTs in self-healing, recent literature about 3D printing, the role of MWCNTs in self-healing, etc).

2. 3D-printed self-healing composite polymer reinforced with carbon nanotubes-based works are already reported. Please clarify more details about the novelty of this work.

3. In Figure 1a, 1b authors need to identify the 0.1, 0.5, 1.0 109 wt.% MWCNTs of filament separately.

4. Author have tried the self-healing study at 80 °C for 60 min. They have show decreasing the depth of the damage from the center part of the hole.

(i)                 Author need to allow more time for showing the complete healing (may be 6 h or 12h)

(ii)              May be increase the self-healing temperature

5. In Figure 2, author need to level the figures of after self-healing for maintain the formats

6. Authors need to be more careful about figure numbers. Figure No. 4 shows Figure 2. So, Figure 4 should be Figure 3.

7. In Figure 4, there is no figure caption (a, b, c, and d) inside the Figure. Please add the caption inside the figure

8. It will be better if the author shows all 0.1, (c) 0.5, and (d) 1 wt.% of 182 MWCNTs Surlyn® specimens along with unreinforced Surlyn® specimen in a single plot. It will helpful to understand, how MWCNTs increase the mechanical properties.

9. In Figure 4 and table 2, it will be better if the author write strain or ε: Elongation in strain(%), ε (%).

10. The author claims mechanical properties increase after the addition of MWCNTs. But why the Surlyn® have a higher Young’s modulus compared with Surlyn® + 1 % MWCNTs, Surlyn® + 0.5 % MWCNTs, and Surlyn® + 0.1 % MWCNTs?

Round 2

Reviewer 2 Report

Comments and Suggestions for Authors

Recommendation: Publish as is; no revisions needed.

Comments:

After carefully reading the revised manuscript and point-by-point response to reviewers' comments, I can fully understand the authors' argument and purpose. Thus, I recommend this paper for publication without further modification.